# Leveraging Sentiment–Topic Analysis for Understanding the Psychological Role of Hype in Emerging Technologies—A Case Study of Electric Vehicles

**DOI:** 10.3390/bs15020137

**Published:** 2025-01-26

**Authors:** Francis Joseph Costello, Cheong Kim

**Affiliations:** 1CUHK Business School, Chinese University of Hong Kong, Hong Kong, China; francisjosephcostello@cuhk.edu.hk; 2Office of Research, aSSIST University, Seoul 03767, Republic of Korea

**Keywords:** hype cycle model, sentiment analysis, topic modeling, psychological impact of technology, electric vehicles, public perception analysis

## Abstract

This study presents a novel approach to examining the psychological impact of emerging technologies through the development of a Hype Cycle Model (HCM), utilizing sentiment analysis and topic modeling. Focusing on electric vehicles, we investigate how public sentiment—captured via social media comments—reflects the psychological effects of technology adoption and hype. Our model integrates both qualitative and quantitative analyses, utilizing sentiment scoring and topic modeling to explore thematic psychological trends. An analysis of approximately 43,000 social media comments on electric vehicles demonstrated that the integration of expert knowledge with public sentiment provides a comprehensive understanding of technology hype dynamics. The results revealed that sentiment analysis enables real-time tracking of emotional responses to emerging technologies, while Correlated Topic Modeling (CTM) offers contextual insights into the positioning of technologies within the HCM. These findings demonstrate that understanding public sentiment towards emerging technologies can provide valuable insights for both organizations and policymakers in technology forecasting and adoption planning. Our approach transforms the traditional black box implementation by Gartner Inc. into a transparent framework that illuminates the psychological underpinnings of technology hype, revealing how collective excitement, expectations, and emotional responses shape the trajectory of emerging technology adoption.

## 1. Introduction

Innovation has become a necessary factor for any institution seeking a world-competitive status ([22]). Forecasting promising innovative technology has become a key responsibility for organizations, where understanding both the technological ecosystem and the psychological impact of innovation adoption is paramount to future success ([61]). Accurately predicting emerging technologies not only provides insight into their performance potential but also helps in understanding the psychological responses and expectations of both users and stakeholders. This psychological dimension is crucial for managing adoption rates and effectively addressing challenges related to change resistance and overestimated expectations ([14]). However, according to a report from McKinsey, the COVID-19 crisis has forced executives to focus on the core aspects of the business in an attempt to sustain operations. Precrisis, executives who viewed innovation as their number one or two priority was over 50%; however, now, this has fallen to just over 20%, showing an apparent decline in innovation focus ([24]). As we navigate these changes, it is increasingly evident that accurate forecasting of promising innovative technologies must now also consider the psychological factors affecting adoption, especially with limited resources available. By incorporating psychological insights—such as how hype influences decision-making—organizations can more effectively manage expectations and mitigate risks associated with overly optimistic projections.

This study seeks to analyze early-stage technology using the Hype Cycle Model (HCM) ([13]; [16]; [39]). The HCM is a well-established model adopted in technology forecasting, capturing not only the technological trajectory but also the psychological responses—such as excitement, disillusionment, and eventual stabilization—that accompany technological adoption. However, two major problems exist: (a) Gartner Inc. does not provide details of how they predict the model, and (b) most studies implementing the use of the HCM do so based on long longitudinal data ([10]; [14]; [28], [29]; [54]; [60]). Consequently, these studies are useful in retrospectively exploring HCM patterns in innovations but do not provide an effective, psychologically informed analysis of early-stage technologies. 

To help rectify this, we implement the use of big data analytics. Big data analytics is an emerging paradigm that has the ability to aid the decision-making processes ([68]) of emerging technology ([35]) and can also capture real-time psychological and emotional responses to new technologies through sentiment analysis and behavioral metrics ([11]). As hype is seen as an overly optimistic human reaction to technology, big data sentiment analysis seems fitting for the task. This notion was also expressed by Fenn and Rakino, who stated that “On the basis of recent improvements in the field of sentiment analysis, which automatically detects the tone of a piece of text from the types of words used (positive or negative), we expect to see a growing availability of tools…But for now the hype cycle remains primarily a qualitative decision tool… (that) still have to rely on the art of expert human judgment” ([16]). Although big data techniques have been explored in the context of emerging technologies, their application for understanding the psychological dimensions of the HCM remains largely unexplored. This integration aims to provide a more comprehensive view of not only the technology’s evolution but also the collective psychological reactions driving its adoption and perception. Based on this notion, we employ a sentiment analysis text mining approach in order to propose an HCM based on the case study of electric vehicles (EV). In order to add rigor, we also employed expert domain knowledge from the EV industry ([32]).

EVs are the latest green trend for many consumers, thanks to companies like TESLA. EVs have a strong reputation as the most viable option for individuals to reduce their CO_2_ emissions ([57]). For this reason, EVs have gained much hype from governments, media, and individual buyers alike. This hype can be seen as a psychological phenomenon fueled by emotional responses such as optimism, enthusiasm, and the fear of missing out (FoMO). Social influence, media coverage, and ambitious targets from governmental institutions contribute to heightened expectations, leading many to believe that the future of EVs is a foregone conclusion. Despite this, protections from local governments, not-fit-for-purpose designs, and questions over its true greenness are still unaddressed concerns ([9]). If we explore the current EV battery technology as an example, research suggests that the extraction of rare raw materials ([8]), size and durability ([5]), and availability of recharging facilities ([32]) remain current issues. This leads us to question whether the hype is restricted to a small subset of influential individuals or whether the nuanced issues surrounding EV technology are being discussed among everyday consumers as well. To explore this, we propose the following research question: Can big data effectively forecast technology adoption and capture the psychological dynamics represented by the Hype Cycle Model?

Our study differs from previous HCM research in three key aspects. First, while existing studies primarily rely on long-term longitudinal data and article counts, we integrate real-time social media sentiment analysis to capture the psychological dynamics of hype. Second, unlike traditional HCM implementations that treat hype as a black box phenomenon, our approach provides empirical evidence of the psychological factors driving technology adoption through sentiment–topic analysis. Third, where previous research has focused solely on technological progression, our approach uniquely combines expert domain knowledge with public sentiment to create a more comprehensive understanding of both the technical and psychological aspects of emerging technology adoption.

Overall, this paper’s main contribution lies in its use of two machine learning techniques to convert the HCM into a more empirical analysis, specifically focusing on the psychological impact of technology adoption. First, we used SentiStrength, a sentiment analysis technique, alongside expert opinions to provide a context-based analysis of the HCM ([65], [67], [66]). Sentiment analysis helps to capture the emotional responses—such as excitement, disillusionment, and renewed interest—that shape the trajectory of technology adoption. Secondly, we implemented an LDA analysis ([6]; [7]). To the best of our knowledge, this is the first implementation of an unsupervised machine learning approach based on a big data social media dataset to analyze the psychological dynamics of hype ([4]; [20]). Furthermore, this study provides a foundation for exploring how sentiment analysis can be used within the HCM framework to understand both the emotional and cognitive responses driving early-stage technology forecasting.

## 2. Literature Review

### 2.1. Hype Cycle Model (HCM)

Till now, studies attempting to interpret a technology’s path through multiple stages of maturity have been inferred through the perception dynamics of the HCM. The value of the HCM’s unique proposition of exploring hype as the main dynamic has gained much traction in the TF and innovation fields ([16]). Increasingly, the use of HCM theory has been utilized in explaining technology such as augmented reality ([39]), environmental change and sustainable technologies ([69]), blockchain technologies ([30]), as well as in medical technologies ([53]). There is utility in the use of HCM for various TF tasks; however, removing the current black box status of the current implementation by Gartner Inc. is a necessary research task.

Originally, the HCM was created to help predict and rationalize early overestimation of technology’s short-term benefits while trying to predict its underestimated longer-term effects ([14]; [16]; [39]). It attempts to explain the general path a technology takes with respect to its expectations or relative maturity in a technological domain ([14]). Importantly, the model also reflects human attitudes and opinions, highlighting the psychological factors and emotional responses that are central to technology adoption ([62]). Since its inception, Gartner Inc. has claimed that the model helps establish the expectations that most technologies will progress through five main phases of maturity: technology trigger, the peak of inflated expectations, disillusionment, the slope of enlightenment, and finally, the productivity plateau (see Figure 1a)—for a detailed explanation, please refer to ([16]). These phases not only reflect technological development but also a collective psyche associated with each stage.

Additionally, the HCM attempts to find consistency between two discrete evolutionary theories: (1) the human-centric hype expectation model (Figure 1a) and (2) the technological S-curve model ([39]) (See Figure 1b). The hype expectation model captures the emotional and psychological journey of stakeholders—ranging from initial excitement to disillusionment—while the S-curve focuses on technological progress. Both models are combined into one single dependent variable alongside time, which becomes the joint independent variable ([3]; [14]). Concerns with this construct have been discussed by [14] ([14]), who underlined how uniting these two distinct models is unachievable. They concluded that the S-curve engineering model could be used to explain technology development, while hype represents psychological fluctuations within this model ([10]; [14]; [25], [26]; [33]; [69]). Although this may be true, concerns about the validity of the technological S-curve model and whether a technology’s trajectory truly follows this S-curve have also been expressed ([15]; [17]; [51]). For these reasons, lingering doubts remain on the viability of Gartner’s HCM model, as well as the conclusions made by Dedehayir and Steinert ([44]; [45]; [48]; [65]).

### 2.2. Previous Attempts in the Literature to Define the Hype Cycle Model

Since the adoption of the HCM into scientific research, the model has gone through various forms of theoretical examination. Predominantly, earlier iterations of the HCM have been centered on adopting a variety of ways to manipulate the *y*-axis. These can be categorized into two broad methods: article/news count and mathematical equations. The most common of these methodologies is the counting of news or scientific articles in order to depict the HCM. Interestingly, Jun et al. used a network and cluster analysis alongside the HCM; however, they did not attempt to theoretically alter the HCM through a data-mining approach ([12]; [36]; [40]; [43]; [56]; [58]; [72]; [73]). 

Till now, professionals and academics alike have yet to truly accept any proposed model for analyzing the HCM, with many conveying doubt about its true nature ([16]). After exhausting a search on prior HCM studies (see Table 1), we identified a major problem as far as context is concerned. Extant literature employs no context within their methodologies (i.e., positive or negative sentiment) and employs long-term datasets in which to evaluate the HCM ([14]). This lack of attention to the psychological dimension—specifically the emotional responses driving hype and disillusionment—limits the model’s ability to provide a nuanced understanding of technology adoption.

Our proposed model seeks to address these limitations by incorporating a context-driven methodology that accounts for emotional responses within the data. By filtering overly positive sentiment (i.e., hype towards a technology), we aim to derive a more accurate prediction that reflects the psychological aspects of technology adoption. Additionally, our model leverages big data within a shorter timeframe, enabling a timelier analysis while maintaining empirical rigor. This approach aims to bridge the gap between the technological and psychological aspects of the HCM, providing deeper insights into the dynamics of hype and the resulting impact on public perception and decision-making.

### 2.3. Sentiment Analysis

Sentiment analysis is an automated or partially automated attempt to find unpredictable and significant patterns in unstructured text ([66]). The fundamental aspect of sentiment analysis is that it is heavily contingent on attempting to find a value in a text’s polarity, usually by identifying the distribution of positive, negative, and neutral sentiments ([16]; [39]). This approach is particularly valuable for understanding the psychological responses of individuals, as it provides insights into collective emotions and attitudes towards emerging technologies. Sentiment analysis has been extensively used in various settings to understand how people emotionally react to different products, technologies, and events ([14]; [25]; [27]; [29]; [31]). 

The adoption of sentiment analysis in this study comes down to two main reasons. As defined by the authors of the HCM, analyzing the hype of technology aims to understand a natural, overly positive reaction to new technology ([65]). Looking at Table 1, most literature thus far has identified metrics that take into consideration a total number of items (i.e., articles, magazines, or news stories). However, a count of the total does not delve into the context of these data sources. This is a big issue that can be solved by sentiment analysis. Deciphering a data set that can be easily separated into positive and negative sentiments can provide a more context-driven analysis of where public opinion is on a given technology ([56]; [71]). Secondly, sentiment analysis allows for an objective analytical approach to separating the data’s sentiment ([64]; [65]). Therefore, sentiment analysis provides a nice complement to extract the potential given the positive ‘hype’ of a target technology ([66]; [70]). Overall, sentiment analysis provides a solution to the methodological problems seen in many articles, as seen in Table 1 ([14]; [16]). 

Although multiple sentiment analysis tools exist, we implemented the use of SentiStrength ([68]). SentiStrength is a tool that performs well in social media texts and has been used in prior social media analysis ([64]; [70]). SentiStrength uses a lexicon-based approach to sentiment analysis and requires no training data to make decisions ([14]; [32]; [69]). It classifies positive and negative sentiments through a dual five-point scheme to classify texts: a score from 1 (weak positivity) to 5 (very strong positivity) and a second score from −1 (weak negativity) to −5 (very strong negativity) ([34]).

### 2.4. The Correlated Topic Model (CTM)

As argued by Kirkels, HCM analysis should be conducted with a mixed-methods approach. Approximating the HCM does not give content-driven awareness of the texts. Thus, it falls short of providing concrete reasons as to why the approximated HCM was devised in its form. To rectify this, we complement the discourse of analysis so as to provide a greater contextual understanding ([21]). They further argue that the combination of both qualitative and quantitative data enhances the strengths of each method, helping to overcome some of the biases seen in the HCM ([7]). We take a similar approach but argue for an analytical approach up till a secondary literature search. Thus, from the same data obtained from social media, we decided to implement a topic modeling approach to help find trending topics of interest that helped form the original HCM analysis. From this, we then performed a secondary literature analysis ([7]). 

Topic modeling allows for context to be extracted from texts based on their importance within the document ([6]). In other words, it describes a document collection by taking advantage of a probabilistic model based on a small number of distributions over a vocabulary. When data are used to fit the model, topics are produced through distributions corresponding to these topics ([2]). Topic modeling has been utilized for the purpose of extracting aspects from unstructured text, and Probabilistic Latent Semantic Analysis (pLSI) ([37]) and Latent Dirichlet Allocation (LDA) have gained the most recognition. 

CTM, proposed herein, is a less well-known model compared with the more well-known implementation of topic modeling based on LDA ([6]; [7]). It follows a computing process identical to the LDA model but does not follow the independence assumption imposed by the Dirichlet in LDA. Rather, it utilizes the logistic norm in order to find correlations between related topics within a dataset. With CTM, it allows for the occurrences of words in other topics and topic graphs, and therefore, the word “car” can be left in the dataset to correlate topics around this main topic based on EVs ([6]). Therefore, with this model, the user can find topics based on correlational relationships of each topic (see Figure 2).

In this generative phase, the CTM differs from the LDA model in that it draws from a logistic norm rather than a Dirichlet. This allows for the model to be more expressive when analyzing document collections where strong correlations may be found between topics. 

## 3. Our Methodological Approach

### 3.1. Technology Focus: Electric Vehicles

EV technology has been identified as the most promising technological solution for the transport sector and thus has attracted widespread hype from practitioners and academics alike. However, concerns around battery technology, the lack of charging options, and the ability of the current grid system to cope with the demand for EVs ([65]) have been raised. With this polarity of opinions on EVs evident in academia, EVs provide a fitting case for understanding the current hype dynamics. Notably, different variations of EVs have already been and failed. The most recent of these—a lead–acid battery configuration—did not last long in the market and was quickly discontinued ([65], [66]). Thus, re-analyzing the newest form of EV technology seems a fitting update through the theoretical lens of the HCM. To undertake our analysis based on EVs, a dataset was utilized from the social media platform YouTube. Approximately 43,000 comments were extracted from videos that were directly about EVs. Only fresh comments were kept that were in the English language. The use of this dataset was because YouTube has a vast amount of data created by the public sphere, which is accessible to any user if wanted. Furthermore, the dataset used in this study was collected between the years 2016 and 2018 and included two and half years of data in total. 

### 3.2. Methodological Justification

Recently, social media data has been extensively regarded by both practitioners and scholars as a voice of the customer as well as the view of the general population ([71]). Therefore, being able to utilize such data can open a great deal of information for whatever the end purpose may be. In this paper, our aim is to predict and understand the HCM and the underlying triggers of why this HCM exists. To this end, we looked at the combination of sentiment analysis and topic modeling. These two tools combined have been presented in many varieties in academia, especially upon researching datasets from social media ([1]; [55]). For this reason, we implement the use of two separate tools (SentiStrength and CTM) to aid our analysis of the HCM. Based on these two quite separate yet seemingly intertwined approaches, we look to view the HCM model through each lens in order to predict the potential hype pattern and then gain insights into the underlying reasons for this predicted HCM. 

### 3.3. Research Phases

Before moving on to each phase of our study’s method and execution, we present our methodological flow. Figure 3 provides a visualization of this with an overarching process view of the approach so that our method can be easily comprehended.

#### 3.3.1. First Phase: Data Collection and Preprocessing

Phase one involved the data extraction stage, including data preprocessing. The search term “electric car” was searched within YouTube’s search bar, and the timescale was set from January 2016 and included videos up till June 2018. This provided a plethora of videos on EVs, from among which 74 were selected based on the criteria of relevance, comment themes, and comment size (total number of comments). Videos were rejected when the video was about EVs, but a famous guest appeared in the video and swayed the comments away from the topic of EVs ([50]). Further, all replies in the corpus collected were removed to eliminate the subjective biases these replies could have. All the comments in our corpus are fresh comments the users created in direct response to the video’s content. As the nature of data acquisition potentially could be subjective or biased, it was necessary to adhere to this criterion to ensure the comments’ context was consistent. Extraction was performed via Google’s Application Programming Interface (API) to collect the comments from the videos selected for the data samples. Additionally, before any analysis took place, the text was preprocessed so that all non-English terms were removed from the sample, as well as stop words and characters that hold no sentiment. Our reasoning for applying a YouTube social media dataset was due to the fact it has had prior success in other literature, for example, ([18]) and ([59]). Additionally, YouTube data were analyzed successfully with the use of the SentiStrength tool by ([64]; [66]).

#### 3.3.2. Second Phase: Sentiment Analysis

Sentiment analysis was performed using SentiStrength, as this algorithm has achieved good results in multiple tasks related to informal social media texts ([64]). The key premise of SentiStrength, compared to most other sentimental analytical tools, is in its extensive libraries of positive words with a given positive score [2~5] and negative words [−2~−5]. On top of this, it takes into consideration spelling errors, boosting words (e.g., very, extremely), negation, as well as repeated letters (e.g., used for emphasis). SentiStrength applies its algorithm separately to each sentence, with each sentence receiving both the most positive and most negative emotion identified. Sentences are split with either a line break in the comment or after punctuation.

To validate the scores obtained from the SentiStrength analysis, we benchmarked it against the trusted and reliable sentiment analysis tool SentiWordNet ([34]). The use of SentiWornNet was not the focus of this paper; however, we wanted to benchmark a sentiment analysis tool that was built around grammatically correct English texts. As the focus of this paper was the use of a social media dataset, SentiWordNet was seen as a good benchmark as it has been shown that it struggles to work when presented with such informal text. Once scores were extracted from both the SentiStrength and SentiWordNet sentiment analysis tools, two researchers with experience with sentiment analysis compared the algorithm’s accuracy manually. Examples of the checks made on each algorithm have been presented in Table 2.

Once the benchmarking had taken place and acceptable results were obtained for analyses, we proceeded with analyzing the data set. Based on the sentiment analysis results, we identified the *y*-axis within the HCM. First, positive comments extracted from SentiStrength were used as the main variable of interest. Neutral and negative comments were included so that we could determine the positive comments’ overall weight within the corpus. Based on all the comments from the YouTube Corpus, $N_{c}$, we consider three possible outcomes from the SentiStrength analysis: total positive comments $C_{P};$ total negative comments $C_{N}$; and total neutral comments $C_{Ne}$. Next, we found the percentage of the total number of positive comments within the total corpus of comments, $C_{\overset{´}{P}}$. This is simply done by dividing all positive comments within the whole corpus. Next, we worked out the average score of the positive comments and multiplied them to find out the final hype score as a percentage, written as $H_{s}$ (formulated as Equation (1)). (1)$$H_{s}=\sum C_{\overset{´}{P}}\frac{\sum C_{P}}{n}$$

In the last step, we must place the positive sentiment score $H_{s}$ within the *y*-axis that represents hype that can sit between 0 and 100. Where *t* is the dataset’s time, and *H* represents a final hype prediction on the *y*-axis based on the social media dataset, formally written in Equation (2). (2)$$H=\int_{0}^{100}\sum H_{s,t}$$

#### 3.3.3. Third Phase: Expert Opinions

To help verify our model, we invited expert reviewers. Two experts were senior researchers within a major automakers’ electric vehicles department, with one researcher from a major academic institution in the department of transport research. The participants were presented with a questionnaire that replicated the model’s principal theory seen in previous work ([34]). Participants underwent a Delphi-style questionnaire that asked them to try and predict the target technology’s stages of the hype cycle. The use of EOs allows for a broader overview of technology before using our model for a timelier analysis. This can then be used to compare and contrast, which helps make a greater analysis. Lastly, prior research has claimed that predicting the HCM without expert domain knowledge is not as effective, and hence, we included it within our analysis.

The Delphi questionnaire was carefully structured to systematically capture expert predictions about EV technology’s progression through the HCM stages. To ensure informed assessments, experts were provided with comprehensive contextual materials, including detailed descriptions of HCM stage characteristics, typical indicators, current technical specifications, and market penetration rates for EVs. The questionnaire was designed to elicit both quantitative predictions and qualitative reasoning about the technology’s progression through each HCM stage. In evaluating each stage of the HCM, experts were asked to make temporal estimations of stage transitions, predict stage durations, and assess hype levels at key transition points using a standardized 0–100 scale. These quantitative assessments were complemented by confidence ratings and detailed explanations of the factors influencing their predictions. The iterative nature of the Delphi process allowed experts to refine their assessments through structured feedback. After each round, participants received anonymized summaries of collective responses, including statistical measures of central tendency and dispersion. This systematic feedback mechanism enabled experts to reconsider their positions considering group responses while maintaining independent judgment. Three rounds of the same questionnaire were conducted until the expert opinions converged. In the second and third rounds of the questionnaire, the experts were also shown prior average scores, and this was used to help them with their responses. Expert opinions turned out to be very fruitful for us to organize a rough picture of the HCM for EVs.

To measure the reliability of these results, we used the Interclass Correlation Coefficient (ICC) to measure the participants’ agreeableness. The ICC is a descriptive statistical tool and a test–retest reliability analysis. Reliability measurement attempts to test data’s true variance over the true variance plus error variance. Therefore, reliability values range between 0 and 1, in which 1 represents the dataset’s greater reliability ([34]). ICC has descriptive power that allows the user to determine the common features within a particular group’s thinking and thus can help reach an intraclass consensus on data that are grouped rather than the paired observational analysis. By doing this, the user can measure both the degree of correlation and agreement of a given group’s dataset ([15]; [17]; [31]; [51]). ICC is represented in Equation (3), where ${MS}_{r}$ equals the mean square for rows, and ${MS}_{e}$ represents the mean square for error. Lastly, ${MS}_{c}$ represents the mean square for columns. All participants had over 5 years of experience in their given position. Given this fact, the two-way, random-effects, absolute agreement model with multiple raters (ICC (2,k)) allowed us to make inferences about the more general population in which those three experts fit ([34]).(3)$$\frac{{MS}_{r}-{MS}_{e}}{{MS}_{r}+\frac{{MS}_{c}-{MS}_{e}}{n}}$$

#### 3.3.4. Fourth Phase: HCM Build-Up

Based on the results from phases two and three, the estimated shape of the HCM for EVs was derived. Based on the EOs obtained in stage three, we had a general idea of where the hype for target technology was. This provided an outline of the HCM based on grounded theory from the expert’s domain knowledge. Although it is not analytically perfect, it provides a good basis for understanding a general time frame. This was also the same for building a general peak and trough of the hype based on the technology. Next, the results from phase two provided analytical precision for predicting the HCM. Analysis of a social media dataset allowed for the convergence of both expert domain knowledge and public opinions. Thus, this combination induces a more validated position on the true reality of hype on the *y*-axis. 

#### 3.3.5. Fifth Phase: Topic Modelling

In order to perform a CTM analysis, we utilized the topicmodels tool within the CRAN R library. Built into the CTM within the topicmodels package is the use of the variational expectation–maximization (VEM) algorithm ([23]; [38]; [46]). Used alongside CTM, the VEM algorithm can be expressed as Equation (4):(4)$$l\left(\mu ,\Sigma ,\beta \right)=log\left(p\left(w|\mu ,\Sigma ,\beta \right)\right)$$

When calculating the estimation in the CTM model, the log-likelihood is calculated with the following $\left(p\left(w|\mu ,\Sigma ,\beta \right)\right)$; however, to control this calculation, it must be transformed into Equation (5):(5)$$log\int \left\{\sum_{z}\left[\prod_{i=1}^{N}p\left(w_{i}|z_{i},\beta \right)p\left(z_{i}|\theta \right)\right]\right\}p\left(\theta |\mu ,\Sigma \right)d\theta$$

After transforming the dataset into a document matrix, we selected the number of topics. Despite the LDA model ([41]; [52]) providing four statistical models for K optimization ([42]), CTM does not boast one. Therefore, we had to rely on trial and error to find the correct K. In our approach, we started with two topics and increased the number of topics up to k = 20. With this approach, we examined all the topic results and found that when k = 7, the most coherent results were presented. Within topics, words include a beta score: a higher beta score shows the significance of this word within the topic in which it is placed ([41]; [47]). Within this study, a threshold of 0.01 beta was used to determine which words to include in our evaluation of a topic. 

## 4. Experimental Results

### 4.1. Study 1

#### 4.1.1. Sentiment Analysis Results

As stated in the second phase, a preliminary check of the SentiStrength algorithm was needed. As can be seen in Table 2, when compared to the SentiWordNet algorithm, SentiStrength identifies the correct sentiment with greater accuracy. However, results from the manual checks also revealed sentences whereby both algorithms were wrong, and thus, we accept that the SentiStrength algorithm is not always 100 percent correct. Nonetheless, the algorithm still performed with good accuracy, and thus, we continued with our study and implemented its use. 

Our sentiment analysis process using SentiStrength involved a dual polarity scoring system where each comment received both a positive (1 to 5) and negative (−1 to −5) score. The final sentiment score for each comment was determined by combining these values. For example, a comment scored 4 (positive) and −1 (negative) would receive a final score of +3, representing strong positive sentiment. The algorithm considers various linguistic elements, including emoticons, boosting words (e.g., ‘very’, ‘extremely’), negations, and punctuation emphasis when assigning scores.

To ensure accuracy, each comment underwent preprocessing to remove noise while retaining sentiment-bearing elements. Comments were then analyzed individually, with SentiStrength evaluating both sentence-level and overall comment sentiment. The mean positive score of 1.346 (Table 3) indicates a moderate level of positive sentiment in the corpus, while the mean negative score of −1.476 suggests similarly moderate negative sentiment. These balanced scores help validate our analysis by showing that the algorithm captured nuanced sentiment rather than extreme polarization. While more recent transformer-based models like BERT and RoBERTa have demonstrated superior performance in some sentiment analysis tasks, we selected SentiStrength for several strategic reasons. First, SentiStrength’s dual polarity scoring system (positive and negative scores simultaneously) is particularly well-suited for analyzing the nuanced nature of technological hype, where mixed sentiments often coexist. Traditional transformer models tend toward more polarized classifications, which could oversimplify the complex emotional responses typically associated with emerging technologies. Second, SentiStrength’s proven track record with social media content, particularly its ability to handle informal language, emoticons, and unconventional spelling—all common in YouTube comments—made it especially appropriate for our dataset. Finally, SentiStrength’s interpretable scoring system allows for a more transparent analysis of sentiment progression over time, which is crucial for mapping sentiment patterns to the HCM’s stages. While transformer models might offer marginally higher accuracy in controlled settings, SentiStrength’s balanced approach to sentiment quantification better serves our specific research objectives of understanding the psychological dynamics of technology adoption.

Next, we set the scale of the HCM’s *y*-axis range from 0 to 100. The maximum positive in our SentiStrength analysis was +4, and the maximum negative was −4. With this, we plot +4 as a hypothetical max of the *y*-axis. This is because, in theory, all the comments could be +4. As seen, the SentiStrength algorithm judges sentences based on positive and negative scores in a combined fashion. Through this score, we obtain a corpus that has the potential to be 100% negative, positive, or even neutral. However, as previously highlighted, the hype is an overly positive sentiment towards technology, and therefore, the total positive sentiment is of interest. Plotting the overall positive sentiment in the corpus as a percentage allowed us to plot the hypothetical peak of the hype. The known positive sentiment of the corpus is then placed on the *y*-axis (between 0–100). Given this logic, we have plotted the model as seen in Figure 4 based on the numbers in Table 3. 

As Table 3 shows, the mean score for the positive sentiment was 1.35. The percentage of these positive comments was calculated to determine how many comments in the total corpus were in the positive range. Knowing the percentage and the mean allowed us to determine the weighted percentage of positive comments (44%, rounded to the nearest whole number). Therefore, to conduct this analysis in relation to the HCM, we posit that this figure of 44% represents the current hype of the EV technology based on the hype cycle’s *y*-axis. 

In our proposed model, positive comments are identified to serve as a basis on which to predict the hype surrounding EV technology. This is because the authors of the HCM suggest that hype’s nature is based on an overly positive, irrational reaction to technology. With the implementation of sentiment analysis, we have systematically identified an area in which to view the HCM and have taken into consideration the context of positivity from the comments. Compared with previous attempts to predict the HCM in academia, our method shows results influenced by this context-driven approach, rather than counting an overall figure seen previously whereby negative and neutral material is included within their datasets.

#### 4.1.2. Expert Opinions

The experts’ first task was to try to predict the number of years in which each stage of the hype cycle fits. Furthermore, we asked them to predict when they believed the positive hype started to increase after the innovation trigger as well as when it started to decrease after the peak of inflated expectations. We also asked the experts to attempt to predict on a scale of [1, 100] how high the peak hype of inflated expectations about EVs would be and how low it would reach in the trough of disillusionment.

A total of three rounds was undertaken to reach a consensus that gave our results statistical significance. The results from the final round, whereby the answers converged the most, are seen in Table 4. Upon verification of the results, we achieved an ICC score of over 75%, confident that the experts had reached an agreeable answer. Table 4 shows the last round results from the expert opinions, and we took the average result of all three experts to plot the *x*- and *y*-axis, as seen in Figure 3. Table 5 shows the summarized results for the ICC results that we achieved based on this last round of results. 

#### 4.1.3. Convergence of Sentiment Analysis and EO: HCM Stage Analysis

As shown in Figure 3, experts agreed that the current form of lithium-ion-powered EV technology had its innovation trigger in the year 2006. Next, the experts decided that the EV technology reached an average peak of 85 on the *y*-axis. Based on the HCM theory, this is when media coverage and hype from the public would have been at their peak positivity. Next, they suggested that this level of hype would fall around 2017, and it is at this point that our sentiment analysis, portrayed as the orange line in Figure 3, meets the expert’s opinions within our model. Based on the comments from the sentiment analysis, 42% were seen to have a positive sentiment. In coherence with the EOs, this provided a grounded position in which to plot our *y*-axis. Considering both the sentiment analysis and expert opinions, their results aligned at similar levels on both the *y*- and *x*-axes around 2017–2018, giving the final hype prediction near the green circle. 

After the year 2018, when all the data were collected for this study, EO analysis was based on a hunch of future outcomes and, therefore, was only used for the visual representation of the estimates as to how EV technology will progress. Thus, our focus remains on the time period leading up to and during the sentiment analysis. As seen in Figure 5, when we compare our results with Gartner’s prediction of EVs within their 2018 report, there is an alignment in the trough of disillusionment. Although not identical, our examination fits in a similar area of the HCM. Thus, the results from our analysis present a methodological approach in which to predict the HCM of a given technology based on analytics. This is in opposition to the prediction seen by Gartner, who does not explain why and how they inserted EV technology into their analysis. This leaves the analytical approach of the Gartner HCM as a black box mystery. 

### 4.2. Study 2

#### Results of the CTM Analysisn

Table 6 shows each topic’s keywords that achieved a beta of 0.01 or more. These identified topics help to shed light on issues that are currently seen within everyday public opinion and can provide information on what needs to be focused on if the technology is to progress through the next stages of the HCM. Although some identified topics seem obvious in our EV analysis, if one were to look at a more obscure technology with the same analytical approach, uncovering unknown factors might prove to be quite prolific in providing new knowledge for a decision-maker to follow up on. Thus, based on the topics’ keywords and identified literature, as shown in Table 6, we created seven observations seen in Table 7.

Through the CTM analysis and a keyword search in academia, Table 7 leads us to three key themes: concerns over batteries, energy source, and price. Although this list is not exhaustive, and many other factors could be at play, this provides a concrete understanding of some of the potentially crucial factors that are limiting EV technology. When coupled with the sentiment analysis whereby a negative outcome was seen, the CTM analysis effectively helps us develop an understanding of the potential reasons why the current EV technology is not being more widely adopted.

## 5. Discussion

As an actor in the decision-making process concerning emerging technologies, deciphering the appropriate timing and applicability of adoption is challenging. In recent years, Technology Forecasting (TF) using the Hype Cycle Model (HCM) has gained increasing popularity but remains subject to criticism due to theoretical flaws ([14]). Previous research has identified that there is no definitive method for numerically operationalizing the *y*-axis, which represents expectations or hype ([63]). As a result, no consensus has emerged regarding how to formulate the HCM model analytically. To address these challenges, this paper presents a novel approach to analytically measure the HCM using machine learning techniques and content analysis, providing insights into the psychological dynamics of the hype surrounding electric vehicles (EVs).

In this paper, we propose a sentimental topic analytical hype prediction model that can be used as a decision support system. Our approach leverages SentiStrength, a sentiment analysis tool, in combination with domain knowledge from expert opinions. The inclusion of sentiment analysis helps to capture the emotional and psychological responses of stakeholders towards emerging technologies, which are crucial drivers of the hype cycle. Additionally, we implemented the Correlated Topic Model (CTM) to provide a more nuanced understanding of the thematic trends and cognitive aspects underlying HCM. By utilizing these tools, our contribution lies in adopting a more context-driven approach that accounts for both the technological and psychological aspects of hype, addressing the gaps highlighted in prior research ([14]). In particular, our approach aims to provide a comprehensive view of both the emotional and psychological dynamics that shape the trajectory of hype. Understanding how individuals react emotionally to emerging technologies—whether through optimism, disillusionment, or renewed interest—is key to refining the HCM and improving its predictive power. In this research, two distinct approaches were employed to answer the proposed research questions: sentiment analysis to capture the psychological tone of public discourse and topic modeling to identify the cognitive themes influencing these emotions. This integration offers a more holistic understanding of the technology adoption process, highlighting the role of psychological factors in driving both hype and realistic expectations.

In study one, we attempt to answer calls from previous researchers who have attempted to explore the model’s dynamics through a multi-level lens. Recommendations for using expert opinions as domain knowledge alongside public/market opinions were proposed as an underdeveloped area of the HCM. With our proposed model, we implemented expert opinion analysis with sentiment analysis of a social media corpus. This is a novel attempt within the HCM field to distinguish between varying sentiments and emotional tones within a given dataset. This differentiation is crucial, as hype is theorized to be an overly positive emotional reaction to new technologies. By specifically extracting and analyzing positive sentiment, our aim of this study was to perform a more context-driven and objective analysis, which elevated insights from the psychological underpinnings of the phenomenon of hype within the HCM.

As demonstrated in our results, we were able to combine both knowledge pools—expert opinion and public sentiment—and form an integrated model. The addition of sentiment analysis allowed for a more timely and analytical approach to understanding the emotional and psychological dynamics driving hype around a given technology. Study One contributes significantly to predicting hype dynamics without separating them into disparate constructs, as previous research often did. By analyzing the interplay between expert knowledge and public emotional responses, we believe that our methodology enhances the robustness and reliability of the HCM. Study Two looked to address the context of the HCM. Through the implementation of a CTM, we extracted statistically relevant words that were related to the case technology. Our results showed that even with little knowledge of a target technology, an actor can start to make some clear inferences about its ecosystem and where it lies on the HCM.

The use of CTM allowed for a greater context-driven analysis of potential reasons as to why the EV technology is situated on the HCM based on sentiment analysis. However, as CTM models only provide words that have a statistical value, interpretations are left up to the user (i.e., Table 7). This could have limitations; however, as [32] ([32]) explains, all decision-making processes are, to some extent, subjective in nature. Overall, the inclusion of the CTM model provides a tool in which a decision can be made with greater insight into the underlying dynamics of technology.

### 5.1. Academic Implications

From a theoretical standpoint, this research’s attempt to expand upon and predict the HCM through big data analysis is a novel attempt within the HCM literature. By implementing sentiment analysis and topic modeling, we have introduced a more context-driven approach that captures the psychological dynamics of hype. Sentiment analysis allowed us to differentiate the emotional tone within the data, specifically identifying instances of overly positive sentiment that represent the hype surrounding a technology. This approach is important because hype is fundamentally a psychological phenomenon characterized by collective emotional responses, including excitement and inflated expectations. By filtering positive sentiment, we were able to begin identifying the overly optimistic perceptions of the technology. Additionally, the use of topic modeling provided insights into why certain sentiments were prevalent, shedding light on the underlying cognitive themes influencing public attitudes towards the technology. As such, we believe this approach has greater precision compared with prior attempts ([14]) in understanding the psychological dimensions of hype. Ultimately, the theoretical implications of this paper suggest expanding the methodological scope of the HCM to include big data and machine learning analysis, with a focus on the psychological and emotional aspects of technology adoption.

### 5.2. Practical Implications

From a practical standpoint, our model provides several practical applications that extend beyond theoretical contributions. First, professionals and policymakers involved in TF should be aware of the new wave of social science in which big data has opened new possibilities. Likewise, the HCM is no exception. Big data allows for the exploration of new phenomena, as well as guides new techniques in decision-making under limited financial resources. As big data is now everywhere, utilizing it can provide a competitive advantage for any firm that is able to accelerate the decision-making process through rigor and best analytic practices ([11]). Addressing the HCM with new methodologies and analytics available in big data, it is possible to accelerate the process of understanding the psychological impact of innovation adoption through hype dynamics of any new technology, which could be crucial to the future success of said technology ([61]).

Second, organizations that need to forecast technology and policymakers can benefit from this approach. Organizations can utilize this framework to monitor real-time consumer sentiment throughout their product development phases, enabling them to identify specific technical concerns before they reach critical mass. This early warning system allows organizations to adjust their research and development investments toward areas of greatest consumer concern while simultaneously informing marketing strategies based on identified topic clusters. For policymakers and government agencies, our findings can be leveraged to design more effective EV (and other innovation technologies) adoption incentives by understanding the underlying psychological barriers to adoption. The sentiment–topic analysis enables the creation of targeted infrastructure development plans that directly address identified consumer pain points. Furthermore, public education campaigns can be crafted to specifically address concerns revealed through topic modeling, while regulatory frameworks can be adjusted based on observed public sentiment patterns.

Third, the integration of sentiment analysis with topic modeling provides organizations with a data-driven approach to understanding both the emotional and rational aspects of consumer behavior. This dual perspective enables more effective strategic planning and resource allocation, which is particularly crucial in the rapidly evolving technology market. Organizations can implement continuous sentiment tracking mechanisms and establish automated systems to identify emerging consumer concerns, ultimately accelerating their technology adoption by addressing both psychological barriers and practical concerns simultaneously.

### 5.3. Limitations

The present study attempted to provide a context-driven exploration of the HCM, but it has limitations. Within our model, some analysis is still left up to the subjective opinions of experts and thus has the potential for bias. Another limitation lies within the choice and implementation of the CTM. Within topic modeling there are various models, and we did not explore all the possible options. Therefore, there may be a limitation in that a performance comparison between the CTM model and other known models, such as LDA, could yield a better analysis. Also, as there is no agreed-upon selection for the number of topics when using CTM, it was left up to us to make that decision, which can inherently have some biases attached to it. Further, all the experts surveyed are within the same geographical area. This has the potential to limit the ICC result’s reliability, but steps were taken to reduce this. Selecting a two-way random-effects model allowed for greater flexibility in selecting randomly from a larger population of those with similar characteristics, as suggested by [34] ([34]).

Lastly, there is a limitation in the use of a big data methodology. Specifically, SentiStrength and CTM have limitations that constitute a further problem. Although both algorithms show high performance in prior academia, accuracies can be questioned. Thus, capturing the underlying hype dynamics could be narrowed with ambiguity in both algorithms present. Future research should aim to replicate and extend this model across various emerging technologies at different stages of development, such as blockchain, virtual/augmented reality, and quantum computing. This will help to evaluate the model’s robustness and applicability in understanding the psychological and emotional dynamics of technology adoption across domains. Also, incorporating deep learning techniques, ensemble models, or hybrid approaches could enhance the analytical rigor, providing richer insights into both technological and psychological aspects of hype and should be explored.

### 5.4. Conclusions

Overall, exploring the HCM model with big data analytics and machine learning was shown to have significant potential to enhance decision-making in technology forecasting. By integrating psychological perspectives, including the emotional responses of individuals, this model provides a more reliable and generalizable framework for understanding the dynamics of hype and technology adoption. These advancements could help professionals, researchers, and policymakers leverage technology to better understand psychological phenomena (including hype) and develop more effective interventions in terms of what issues still need resolving and provide guidance on where more investment is needed so that the public expectations towards the technology are realized.

## Figures and Tables

**Figure 1 behavsci-15-00137-f001:**
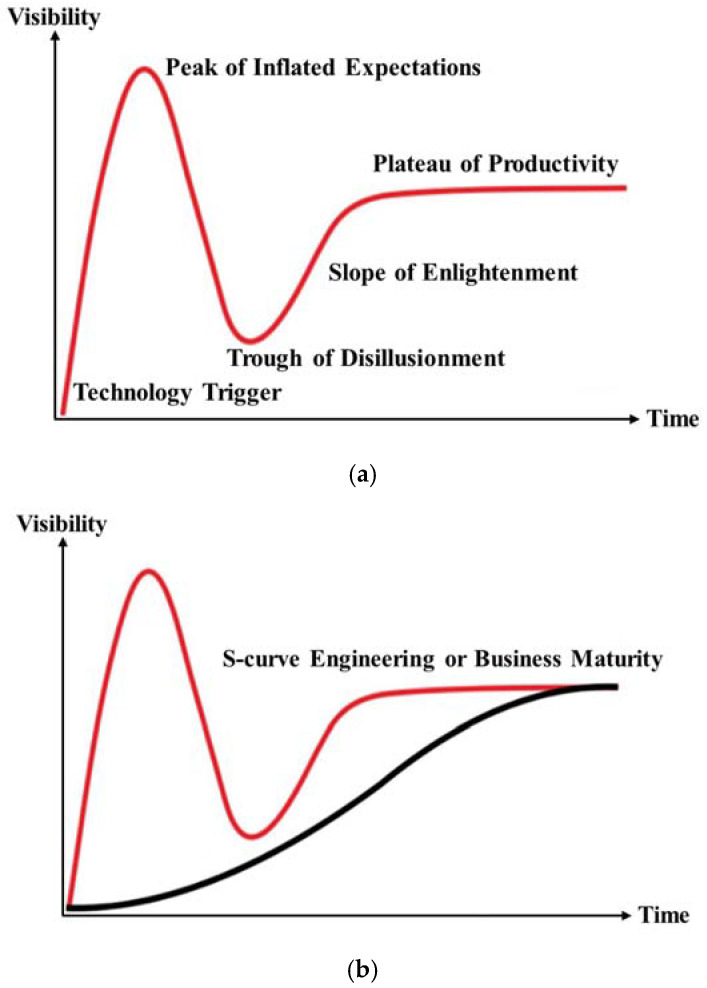
(**a**) The hype expectations curve. (**b**) The hype cycle model.

**Figure 2 behavsci-15-00137-f002:**
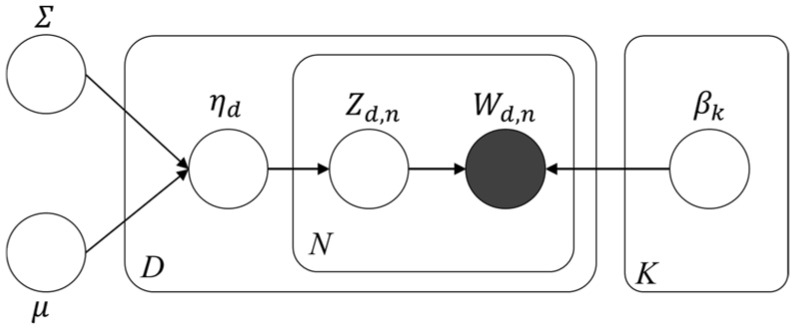
The correlated topic model ([6]).

**Figure 3 behavsci-15-00137-f003:**
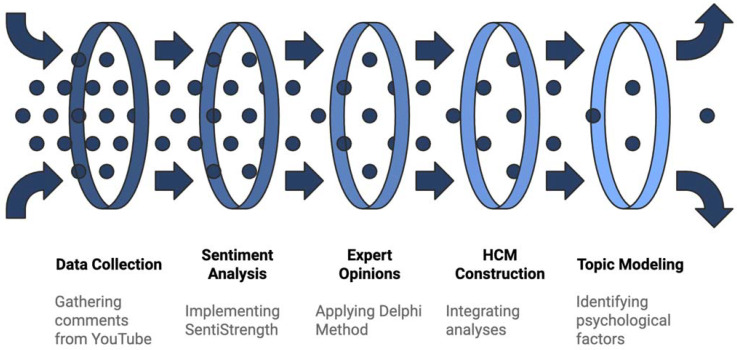
An overview of our method.

**Figure 4 behavsci-15-00137-f004:**
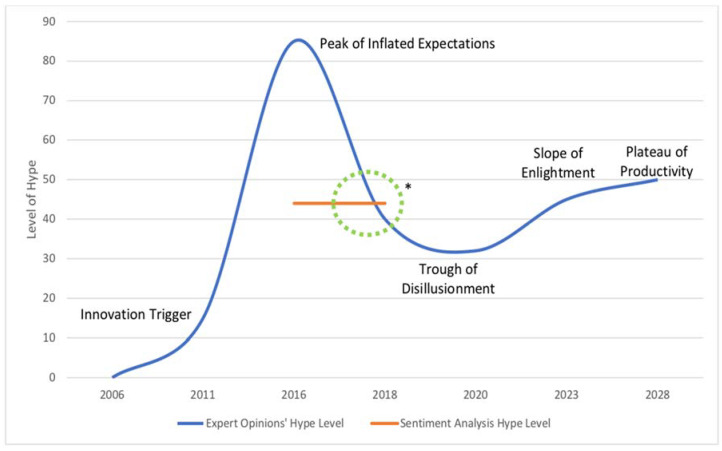
Predicted Hype Cycle Model for EVs. Note: The orange line represents the sentiment analysis results, as seen in Table 3. * The orange line represents the sentiment analysis results as seen in Table 3.

**Figure 5 behavsci-15-00137-f005:**
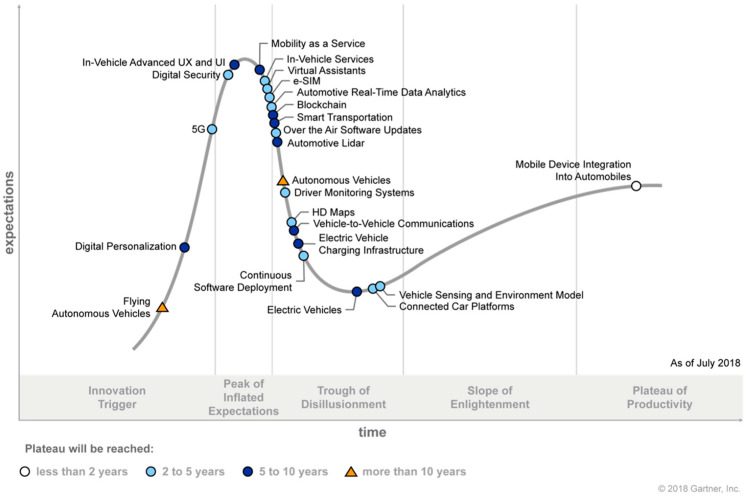
Gartner’s HCM for connected vehicles (including EVs).

**Table 1 behavsci-15-00137-t001:** Previous studies on HCM (last ten years).

Author	Context	Data Source	*y*-Axis	*x*-Axis (Years)	Data Mining
[54] ([54])	Stationary fuel cell	Newspapers	Article count	15	No
[26] ([26])	Hybrid cars	Google Trends	Search traffic	8	No
[25] ([25])	Hybrid cars	Google Trends	Search traffic	8	No
[3] ([3])	Biofuels, hydrogen, natural gas	Newspaper, Journals	Article count	8	No
[33] ([33])	Fuel cells	Newspapers	Article count	15	No
[49] ([49])	Neuroimaging	Journals	Article count	21	No
[69] ([69])	VoIP, gene therapy, high-temp superconductivity	Newspapers	Article count	11~14	No
[29] ([29])	Hybrid cars	Google trends	Search traffic	8	No
[10] ([10])	OLED	Journal, Patents	Math formulae	23	No
[32] ([32])	Biomass gasification	Conferences, Journals	Article count	15~18	No
[28] ([28])	DVR, fridge, fishing rods, beds, alcohol, bottled water	Google Trends, sales	Search traffic, sales	8~15	No
[14] ([14])	Tidal power, IGCC, solar panel	Google news	Search traffic	5~6	No
[60] ([60])	OLED, LCD, superconductivity	Journals	Rate equation	20~32	No
[27] ([27])	Google trends	Journals	Network analysis	11	Yes

**Table 2 behavsci-15-00137-t002:** Comparison of the SentiStrength and the benchmarked SentiWordNet algorithm.

YouTube Comments	SentiStrength				SentiWordNet	
	+	−	=	Sentiment	Score	Label
‘great video it mentions all the things people miss when they state hydrogen as better induction motors are quite simply far more efficient and better than any combustion engine’	3	−1	2	Fairly positive	0.00967	Neutral
‘oil mafia is not gonna move away without fight so we are seeing lot of trash talking against elon musk and tesla’	1	−3	−2	Fairly negative	0.05368	Positive
‘single speed transmission sounds cool but it makes me sad that i wont be listening my car changing gears when accelerating or even’	2	−2	0	Neutral	−0.03568	Negative
‘im kinda starting to like evs now the range is even better and there are a few models that look more like an suv than the bolt i just cant afford a new one yet even with the tax credit they dont compare in price to economy cars’	1	−1	0	Neutral	0.04890	Positive
‘dough i think gm is trying to scare all the electric car market consumer by putting so much cool stuff in its first ev car’	2	−4	−2	Negative	0.00875	Neutral

**Table 3 behavsci-15-00137-t003:** Empirical results from SentiStrength analysis and calculations (hype *y*-axis prediction).

SentiStrength Score	$N_{C}$	Mean Scores	$C_{\overset{´}{P}}$	$H_{s}$	H
Positive comments (+0.1~+4)	14,157	$C_{P}-$1.346	32.67	0.4401	44
Negative comments (−0.1~−4)	8312	$C_{N}-$−1.476	19.18	−0.2834	−28
Neutral comments (0)	20,863	$C_{Ne}-$0	48.15	0	0

**Table 4 behavsci-15-00137-t004:** Last round results from the expert opinions based on a Delphi method.

	Stage 1: Innovation Trigger	Time Hype Starts to Build	Stage 2: Peak of Inflated Expectations	Stage 2: *y*-Axis Prediction	Time Hype Starts to Fall	Stage 3: Trough of Disillusionment	Stage 3: *y*-Axis Prediction	Stage 4: Slope of Enlightenment	Stage 5: Plateau of Productivity
Exp 1	2010	2014	2020	100	2021	2021	20	2024	2030
Exp 2	2002	2009	2013	70	2017	2020	40	2024	2028
Exp 3	2005	2009	2015	85	2017	2020	35	2022	2026
Average	2006	2011	2016	85	2018	2020	32	2023	2028
Max	2010	2014	2020	100	2021	2021	40	2024	2030
Min	2002	2009	2009	70	2017	2020	20	2022	2026

**Table 5 behavsci-15-00137-t005:** Empirical results from the ICC analysis of expert opinions.

	Type	Score	F	Df1	Df2	P	L-Bound	U-Bound
*x*-axis prediction	ICC2k	0.86	7.0	9	18	0.00024	0.51	0.95

**Table 6 behavsci-15-00137-t006:** Results from the correlated topic model (k = 7).

Topic	Term	Beta	Topic	Term	Beta	Topic	Term	Beta
1	drive	0.0253	3	will	0.0278	6	tesla	0.0933
1	motor	0.0216	3	mile	0.0252	6	like	0.0476
1	now	0.0187	3	rang	0.0221	6	look	0.0348
1	way	0.0174	3	model	0.0200	6	bolt	0.0169
1	engine	0.0166	3	price	0.0176	6	model	0.0158
1	world	0.0113	3	replac	0.0126	6	make	0.0122
1	gear	0.0110	4	love	0.0149	6	chevi	0.0114
1	speed	0.0110	4	great	0.0142	6	sell	0.0103
1	make	0.0106	4	know	0.0141	6	realli	0.0102
2	battery	0.0417	4	thank	0.0126	7	just	0.0364
2	can	0.0372	4	elon	0.0105	7	car	0.0361
2	charg	0.0339	5	power	0.0351	7	get	0.0226
2	will	0.0325	5	energi	0.0269	7	dont	0.0225
2	car	0.0302	5	car	0.0210	7	work	0.0172
2	one	0.0301	5	use	0.0175	7	like	0.0165
2	peopl	0.0251	5	fuel	0.0172	7	much	0.0154
2	buy	0.0237	5	oil	0.0171	7	good	0.0154
2	time	0.0232	5	solar	0.0156	7	new	0.0151
2	get	0.0215	5	coal	0.0140	7	futur	0.0139
3	cost	0.0339	5	produc	0.0116			
3	year	0.0284	5	effici	0.0116			

**Table 7 behavsci-15-00137-t007:** Statements made on the electric vehicle technology based on the CTM results.

Topic	Topics Based on CTM and Keyword Search in Academia
1	People are enthusiastic about the use of electric motors as the energy used during their use is a clean energy source compared to diesel or gasoline. However, people are still interested in traditionally powered engines that have gears ([19]).
2	If the battery charge time is reduced, people will be more willing to buy EVs ([52]).
3	People are not only concerned with just the initial price of EVs but also the price of battery replacement. Furthermore, people are still concerned with the range of the battery from fully charged ([3]; [14]).
4	Due to TESLA’s CEO Elon Musk, people have started to have an interest in EVs ([32]; [69]).
5	People are still concerned about the source of electricity production (for example, solar or oil) when it comes to EVs ([16]).
6	Amongst all the EV models, TESLA is the brand that is seen the most by consumers within media ([23]; [38]; [46]).
7	Many people do not want to buy EVs just because they are new; however, many people like them and think they are the future of private transportation.

## Data Availability

Data can be shared upon request.

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
