# Peer review of "Leveraging Sentiment–Topic Analysis for Understanding the Psychological Role of Hype in Emerging Technologies—A Case Study of Electric Vehicles"

_behavsci, 2025, doi:10.3390/bs15020137_

Round 1

Reviewer 1 Report

Comments and Suggestions for Authors

Limitations of Existing Models: The study highlights that existing models, such as those from Gartner Inc., lack detailed predictive capabilities and primarily rely on long-term data, which may not effectively analyze early-stage technologies .

Role of Big Data: The authors emphasize the potential of big data to enhance decision-making by capturing real-time psychological and emotional responses to new technologies through sentiment analysis and behavioral metrics. This approach aims to provide a more comprehensive understanding of technology adoption .

Sentiment Analysis and Topic Modeling: The research combines sentiment analysis and topic modeling to predict and understand the HCM and the underlying triggers for its existence. This dual approach is particularly relevant for analyzing social media datasets .

Data Collection Methodology: The study describes a rigorous data collection process, where comments from YouTube videos about electric vehicles were analyzed. The selection criteria ensured that the comments were relevant and directly related to the topic of EVs, minimizing biases .

Expert Opinions: The inclusion of expert opinions is noted as a critical component of the research, providing domain knowledge that enriches the analysis of the HCM and the technology being studied .

Reviewer 2 Report

Comments and Suggestions for Authors

With the development of self-driving cars using artificial intelligence and secondary battery technology, the interest in electric vehicles is high. This study summarizes the sentimental factors and several topics about electric vehicles in the public using text mining techniques. This study is very timely and the topic is interesting. The paper is well written, with a logical progression and easy to understand. However, as a descriptive study, the practical implications are important rather than the theoretical rigor of the study. Therefore, the practical contribution of the study could be strengthened with specific references to how the results of this study can be utilized in practice. 

Reviewer 3 Report

Comments and Suggestions for Authors

[General comments]

1. A more detailed explanation of the scoring process and results in the sentimental mining analysis is needed.

2. Please add the differentiation of your research in the introduction.

3. The research method is evaluated as innovative. Please visualize and present the research composition.

4. In the conclusion section, it is necessary to emphasize this study's expected effects and contributions.

[Detailed comments]

1) The authors used SentiStrength for sentiment analysis, a well-established tool known for its simplicity and effectiveness in various contexts. However, it would be beneficial to understand the rationale behind selecting SentiStrength over more modern techniques such as BERT, RoBERTa, or other transformer-based models. These contemporary methods perform superiorly in capturing nuanced sentiments and handling complex linguistic structures. Providing a comparison or justification for the choice of SentiStrength would strengthen the study's methodological rigor and offer readers a clearer understanding of the trade-offs involved.

2) The Delphi study is a crucial component of the research, and readers need to understand the materials presented to the experts. Could the authors please elaborate on the specific items or information shown to the experts during the Delphi study? This includes any questionnaires, data sets, or other materials used to solicit their opinions. Providing this information will enhance the transparency of the study and allow readers to better evaluate the findings and the process by which consensus was reached.

Round 2

Reviewer 1 Report

Comments and Suggestions for Authors

Author answered the comments nicely and incorporated all the concerns raised.

Author Response

Comment 1: Author answered the comments nicely and incorporated all the concerns raised.

Answer: We do appreciate valuable feedback from the reviewer, and we are very happy that the reviewer is satisfied with the revised version.

Thank you!